# Triggering of the immune response to MCF7 cell line using conjugated antibody with bacterial antigens: In-vitro and in-vivo study

**Mohammad Khosravi**[1]*, **Kaveh Khazaeil**[2], **Fatemeh KhademiMoghadam**[3]

**1** Department of Pathobiology, Faculty of Veterinary Medicine, Shahid Chamran University of Ahvaz, Ahvaz, Iran, **2** Department of Basic sciences, Faculty of Veterinary Medicine, Shahid Chamran University of Ahvaz, Ahvaz, Iran, **3** Department of Biology, Faculty of Science, Shahid Chamran University of Ahvaz, Ahvaz, Iran

* m.khosravi@scu.ac.ir

**Data Availability Statement:** All relevant data are within the paper and its Supporting Information files.

## Abstract

The current study intended to trigger the immune response to cancer cells by using antibodies conjugated with bacterial antigens. The protein membrane of the MCF7 cell line was extracted and specific antibodies against cell membrane antigens was produced in rabbits. The specific antibodies were purified using chromatography methods and linked to *E. coli* antigens or doxorubicin using Diethylenetriamine pentaacetate (DTPA) linker. After confirmation of the conjugation process using SDS-PAGE and ATR-FTIR methods, the MCF7 and HUVEC cells were treated with various concentrations of the prepared conjugated antibodies along with human serum. The toxicity of each treatment against MCF7 and HUVEC cells was evaluated using the MTT assay. Also, polylactic acid scaffolds that contain $10 \times 10^4$ MCF7 cells were surgically placed in the peritoneal cavity of the rats. After treatment of each group, induction of the inflammatory responses was evaluated on stained histological sections of the scaffolds. The lowest cytotoxic doses of the antigen conjugated-antibody, doxorubicin-conjugated-antibody was 4 and 1 μg/mL, respectively. Doxorubicin conjugated antibodies displayed greater toxicity on both MCF7 and HUVEC cells. The in vivo finding revealed that the inflammatory cells were significantly higher in treating animals with antigen conjugated-antibody. The current synthetic agent stimulated the serum toxicity and induced an inflammatory response to MCF7 cell lines. Targeting of the bacterial antigens on tumor sites by immune system elements, could limit the growth of the tumor cells.

## Introduction

An annual increase in the death rate due to various types' of the cancers is observed and females are the most affected population by breast cancer [1]. To study breast cancer, the human breast cancer cell line, Michigan Cancer Foundation-7 (MCF7), is often used in experimental studies because the mammary epithelium of the breast cancer patient, like the MCF7 cell line, has individual characteristics. Therefore, this cell line is a widespread investigative tool in breast cancer research [2]. The conventional treatments like surgical resection,

**Funding:** Our research project was financially supported by Shahid Chamran University of Ahvaz, Iran, under the grant number of 98/3/05/14909. The funders had no role in study design, data collection and analysis, decision to publish, or preparation of the manuscript.

**Competing interests:** The authors have declared that no competing interests exist.

radiotherapy and chemotherapy are shown to be ineffective in some cancer patients; thus, some alternative options of target therapy have been developed in recent decades. However, due to the low rate of efficacy, feasibility, availability, specificity and selectivity, the number of other approaches is limited to the methods like photodynamic therapy, gene therapy, hyperthermia therapy, diet therapy, insulin potentiating therapy and bacterial treatment [1].

Several antibodies were prepared to help target the cancer cells. In fact, what they do is to target the cell surface antigens by using modified antibodies to limit the cancer's growth by multiple mechanisms [3]. Antibodies can modulate the immune response and inhibit tumor cell signaling [4]. Anti-cancer antibodies that target cell surface antigens have low cytotoxic effects; so, they need toxic agents for the eradication of the tumor mass. The developed new conjugated antibodies with natural agents, radioisotopes or chemotherapeutic agents have suitable efficacy in several hematological cancers [5]. Also, unconjugated antibodies have been used in non-leukemic cancer, such as breast tumors [6]. Antibodies could target the tumor niches, angiogenesis factors, tumor growth factors and tumor ligands; these related strategies should enhance the anti-tumor immune responses [7]. Also, antibodies activate phagocytosis, natural killer (NK) cells and classical pathways of the complement system. In spite of the various mechanisms, most of the antibodies do not have a sufficient cytotoxic effect on cancer cells [8]. Antibody-Antigen-Adjuvant conjugates motivate more CTL responses and anti-tumor immunity than antigen-adjuvant conjugates in animal models; this approach was suggested for obtaining a powerful vaccination response [9]. The conjugation of the dual antibodies against cancer cells and T cell markers to protein nanoparticles is used for the redirection and activation of the T lymphocytes to cancer niches [10]. Decreasing the drug side effects by targeting delivery of the therapeutic agents is the main objective of cancer therapy; these strategies are mostly dependent on specific recognition of the targets by using antibodies. But there are many limitations to developing anti-cancer antibodies [11]. The related problems are lack of efficacy, low cytotoxicity and cellular uptake, the use of potent toxic molecules like tubulin inhibitors, DNA intercalators, pyrrolobenzodiazepines, crossing reactions with normal cells, which are known as off-target effects, and the synthesis of a heterogeneous mixture of the conjugated antibodies [12, 13]. Another important issue includes the methods of conjugation, which determines the activity and potency of the conjugated antibodies; release of the toxin or drug inside the cancerous tissues occurs by cleavage of the linkers. Currently, several smart drug delivery and release mechanisms have been explored that elevate the effectiveness and reduce the destructive effects of the anti-cancer agents [14]. The cancer antigens often share similar epitopes with normal cells and high levels of immunosuppressive cytokines; as a result, they can evade the host immune response. Here, enhancement the cancer antigenicity and triggering the immune response appears to help the host immune system to target the malignant tumors [15]. The clinical outcome of the cancer patients is related to the local proinflammatory or suppressor cytokine ratios [16]. The cytokines, which are involved in innate immune cell recruitment and activation, play main roles in modulating the immune response. Inflammatory reactions stimulate the production of the reactive oxygen species, proteases, and other derivative enzymes, which can directly disrupt the cancer cells [17].

In the meanwhile, stimulation of the cellular immune response is another aspect of the inflammatory response induction [15]. The objective of study is evaluate the in vivo and in vitro stimulation of the inflammatory response and anticancer effects of the conjugated antibodies to *E. coli* antigens. Recognition of the bacterial antigens by components including humoral elements and cellular parts of the immune system on the surface of the tumor cells could enhance cytotoxicity against the targeted tumor cell by specific antibodies. Additionally, doxorubicin as the most commonly chemotherapeutic drugs, in antibody-conjugated and unconjugated forms, were used as controls.

## Materials and methods

### Materials

Two main materials used in this experiment, polylactic acid nanofiber scaffold, were purchased from Sigma. The human umbilical vein endothelial cells (HUVEC) and Michigan Cancer Foundation-7 (MCF7) were obtained from Iran Pasteur Institute, Tehran. The culture media was Dulbecco's Modified Eagle's Medium (DMEM) (Gibco, 11965118) supplemented with 10% (V/V) heat-inactivated fetal bovine serum (FBS) (Gibco, 10082139), 100 U/mL penicillin and 100 μg/mL streptomycin (Gibco, 15140148) or 0.25% trypsin (Gibco, 25200056). Other reagents including 3-(4,5-dimethylthiazol-2-yl)-diphenyltetrazolium bromide (MTT, M2128), sepharose 4B (Sigma–Aldrich, Product Number: 4B200), diethylaminoethyl cellulose (DEAE-C) (Sigma, Number: D3764) were purchased from Sigma Company. The HRP-conjugated anti-rabbit IgG was supplied from Immuno Chemistry Technologies company, USA (HRP AffiPure Goat anti-Rabbit IgG Fc, Catalog Number: 6293).

### Cell culture

MCF-7 and HUVEC cells were grown in the Dulbecco's modified eagle medium (DMEM) supplemented with 10% FBS and 1% penicillin-streptomycin. Then, the cells were grown in 25 or 50 $cm^2$ culture flasks and maintained at 37˚C and 5% $CO_2$ throughout the study; the prepared cells were harvested after attaining confluence by trypsinization. Finally, following centrifugation (1500 g for 6 min), cells were suspended in the DMEM medium and used in the following steps in the study.

### Preparation of the MCF-7 cell surface membrane

The membrane of MCF7 cells was extracted according to Liu et al. [18]. The lysis buffer was prepared using a mixture of 100 mM Tris-HCl (pH 7.5), 1 mM EDTA, 2% Triton X-114 and 1% protease inhibitor mixture (10 mg/mL PMSF). In brief, $1 \times 10^7$ cells were mixed and incubated with 1 mL cold lysis buffer for 15 min at room temperature. The mixture was centrifuged at 13,000 rpm for 20 min and the supernatant was collected carefully; an equal volume of sucrose cushion, which contained 6% sucrose, 150 mM NaCl, 0.06% Triton X-114 and 10 mM Tris-HCl, were added to the prepared supernatant and placed in 37˚C incubator for 5 min. Then, a 9-fold volume of precooled acetone was added to the precipitated part. After an overnight incubation at -20˚C, the membrane protein was collected by centrifugation at 13,000 rpm for 20 min, and resuspended in 1 mL of deionized water. The concentration of the membrane protein was determined by the Bradford assay.

### Ethics statement

All animal procedures were carried out under the terms of a project license issued by the animal ethics committee of the veterinary faculty of Shahid Chamran University of Ahvaz, Iran. The animals were kept under controlled conditions at 22˚C and 50% humidity; they had ad libitum access to food and water during maintenance. To adapt the animals to the keeping conditions, one week before the start of the in vivo experiments, they were placed in environmental conditions. During treatments, the used animals, including rabbits and rats, were anesthetized with intramuscular or intraperitoneal injection of a mixture of ketamine-xylazine. After anesthesia, taking blood samples or a surgical operation is performed. After surgery, the rats were kept in separate cages.

## Immunization

Two male White New Zealand rabbits with weights of 2±0.2 Kg were maintained in an animal cage according to the animal care division. The cell surface membrane, at a concentration adjusted to 250 μg in 0.5 mL, mixed with an equivalent amount of the complete Freund's adjuvant. The prepared antigens were injected subcutaneously and intramuscularly to each rabbit. The booster antigens were prepared by mixing 0.5 mL of the cell surface membrane at a concentration adjusted to 125 μg with an equal amount of the incomplete Freund's adjuvant. However, the prepared antigen was injected as before. An in-house ELISA test was designed and used to assess the antibody titer of each immunized rabbit in the usual route. The immunized rabbits were anesthetized with an intramuscular injection of ketamine-xylazine 25:5 mg/mL at 1 mL/kg body weight and the blood samples were collected; the hyperimmune sera were harvested and stored at -20°C.

## Antibody purification

The total IgG of the hyperimmune sera was purified by using ion exchange chromatography on DEAE-C column (Sigma, D376) according to guidelines [19]. Purification of the specific antibodies was done using affinity chromatography. Briefly, MCF-7 membrane protein, 5 mg was coupled to sepharose 4B (Sigma, 4B200) by the cyanogen bromide linker. The extracted total IgG was incubated with activated sepharose beads for 60 min. The beads were washed several times with PBS and were transferred into a column. The specific antibodies were released from the column using glycine buffer pH 2.5 and immediately neutralized using Tris buffer pH 9.5. The crossing reactive antibodies were absorbed by mixing the purified antibodies with HUVEC cells and centrifugation, the supernatant was taken as specific anti-MCF-7 antibodies. The quantity of the purified antibody was tested using the Bradford protein assay. The purified antibodies were evaluated using sodium dodecyl sulfate polyacrylamide gel electrophoresis (SDS-PAGE).

## Coupling of the bacterial antigen and doxorubicin to antibodies

An in-house synthesis procedure was carried out to conjugate the antibodies to bacterial antigens. The antibodies 1 mL (1 mg/mL) was mixed with Tris-HCl 0.1 M, pH 8, which contained DTPA 200 mg and incubated overnight at room temperature. After dialysis for 24 h at 4°C sonicated *E. coli* antigens (100 μg) or doxorubicin (100 μg) were added to the activated antibodies. Samples were shaken occasionally to prevent bead sedimentation for 30 minutes. After the addition of glutaraldehyde 8 mM at room temperature, the beads were shaken for the next 2 h and incubated overnight at 4°C. Un-reacted sites of the antibodies were blocked using 4 mg/mL skimmed milk. The mixture was incubated for 2 hours at room temperature and was dialyzed for 48 h in phosphate buffered saline at 4°C. The prepared conjugated molecules were dispersed in a filtered storage buffer which included sterilized PBS containing BSA 1%. The produced conjugates were aliquoted and stored at 4°C.

## Attenuated Total Reflection Fourier-Transform Infrared Spectroscopy (ATR-FTIR) analysis

The prepared antibodies conjugated to doxorubicin and bacterial antigens, purified anti-MCF7 antibodies and doxorubicin in phosphate buffered saline (PBS) were analyzed using ATR-FTIR spectroscopy. The potassium bromide pellet method was used in ATR-FTIR and the spectrum was obtained using 16 scans with the absorption range of 4,000 to 400 cm$^{-1}$.

## In vitro cytotoxicity test

The MCF7 and HUVEC cells were grown in 50 cm$^2$ cell culture flasks in DMEM medium, supplemented with 10% FBS, 100 U/mL penicillin and 100 U/mL streptomycin; the medium was changed for 2 days intervals. The cells were trypsinized and after being washed were seeded into sterilized 96 wells microtiter plate (10.000 cells/well) and placed at 37˚C incubator (5% CO$_2$). After 24 hours, the wells were washed with PBS, and treated for 2 h with 100 μl of DMEM medium which contained different concentrations (3, 4, 10 and 40 μl) of antibody conjugated to *E. coli* antigens with or without human sera, antibody conjugated to doxorubicin, doxorubicin, antibody and various concentrations of human sera. After washing off the wells, thiazolyl blue tetrazolium bromide solution (MTT) 100 μl of 0.5 mg mL$^{-1}$ in culture media was poured into the wells, and incubated for 3 h at 37˚C in a CO$_2$ incubator. Dimethyl sulfoxide (100 μL) was added to the wells and the plate was incubated for 30 minutes. The absorbance of the plate was measured at 600 nm with a microplate reader (AccuReader, Taiwan). Also, the treated cells were analyzed using the usual morphological assay [20, 21]. The percentages of cell viability were used to determine the IC50 values, which is the concentration of drugs inhibiting 50% of the cell growth compared with that of the untreated control cultures.

## Scanning electron microscopy characterization

The MCF7 cells were placed on the prepared polylactic acid (PLA) scaffold specimens (5 mm in diameter, sterilized with UV and coated with 1% matrigel) with a final seeding density of 10×10$^4$ cells/scaffold in 96 well plates and incubated in DMEM for 6 days. Thereafter, the scaffolds were fixed with 2.5% glutaraldehyde in phosphate buffer saline (PBS) solution for one hour at room temperature. The fixed scaffolds were washed with PBS and dehydrated through a graded series of ethanol. The MCF7 cell attachment to the processed nanofiber PLA scaffolds was analyzed using scanning electron microscopy (SEM).

## In vivo cell cytotoxicity test

The 18 four-week-old female rats were housed and acclimatized for 7 days before starting the experiments. During experimental periods, the rats were housed in separate cages. The rats were fed with rodent pellets and water ad libitum. All rats were anesthetized with intraperitoneal injection of a mixture of ketamine and xylazine (25:5 mg/mL at 1 mL/kg body weight). The incisions were made on the middle of the abdomen 1 cm in diameter. The prepared PLA scaffolds which contain 10×10$^4$ cells/scaffold were washed and placed on the peritoneal cavity of the rats and the incision site was stitched.

After 48 hours of surgery, rats were divided into three groups of control, Ab-Ag and Ab-Dox; the grafted rats received daily 100 μL of filtered phosphate buffered saline, antibodies conjugated to *E. coli* antigens (100 μg/mL) and antibody conjugated to doxorubicin (10 μg/mL) for seven days, respectively. The treated animals were euthanized by overdose of intraperitoneal injection of sodium pentobarbital (800 mg/Kg of body weight). The nanofiber scaffolds were removed from euthanized animals and washed with sterilized phosphate buffered saline and fixed immediately in 10% buffered formalin, dehydrated, and embedded in paraffin blocks. Seven micrometer paraffin sections were stained with hematoxylin–eosin (H&E) stain for histological observations using standard protocols [22]. Histological analysis was carried out using a light microscope (Olympus, 1 51, USA) and image tool software (version 3). The inflammatory and fibroblast cell density and cell morphology were determined on stained sections. For determining cell density, a random sample of recovered images (1.0 mm2) was taken in high magnification (400 and 1,000), and nuclei of the cells were counted by image

tool software. The results were an average of at least six specimens. The cell number/area ratios were counted for each section and the results were compared between the groups.

## Statistical analysis

In the MTT assay, each concentration was assayed in 4 wells (n = 4) and repeated in three independent experiments. Statistical analyses were performed by SPSS statistical software (version 20.0, SPSS). Values with $P < 0.05$ were considered as statistically significant. Data was analyzed using one-way ANOVA and expressed as mean ±SD.

# Results

## Coupling of the bacterial antigen and doxorubicin to antibodies

The *Escherichia coli* sonicated antigens or doxorubicin was coupled with anti-MCF7 antibodies; this step was successfully confirmed by using SDS-PAGE (Fig 1), and ATR-FTIR analysis (Fig 2).

Compression to an alone antibody, conjugation of the bacterial antigens to the antibody molecules formed several new FTIR peaks at 1061–1467 cm$^{-1}$. The antibody-doxorubicin complex showed a different FTIR pattern from antibody or doxorubicin alone; the blue shifted bands are in the range of 1061–1638 cm$^{-1}$; also, three bands at 2966, 2923 and 2851 cm$^{-1}$ were disappeared in the produced doxorubicin-antibody complex (Fig 2). The mentioned absorption peaks in the FTIR spectrum of B–Ab and Dox-Ab indicated conjugation between the reagents. The obtained bonds at 3441 cm$^{-1}$ is related to hydroxyl and amine groups. The peak around 1638 cm$^{-1}$ reflected the C-O bond and also antibody amide I; and the peak at 1467 cm$^{-1}$ was assigned to the N–H bond of antibody amide II and H2O; these carboxyl and hydroxyl groups could interact in the coupling process.

## In vitro effects of the prepared anti-cancer agents

The MCF7 and HUVEC cells were treated with different concentrations of polyclonal anti MCF7 antibody (Ab), antibody conjugated to *E. coli* antigens (Ab-Ag), doxorubicin (Dox) and antibody conjugated to doxorubicin (Ab-Dox). The cytotoxicity of synthesized anticancer agents on MCF-7 and HUVEC cells was determined by MTT assay; results are presented in following. In brief, the in vitro results showed excellent repeatability and that the optical absorbance of each well was similar in the other three parallel wells and the three-time- repeated experiments. The obtained results indicated the dose dependent effects of the prepared anticancer agents; the cell viability rates decreased significantly ($P < 0.05$) with increasing concentrations of all the tested agents. Doxorubicin was more toxic against both MCF7 and HUVEC than antibodies conjugated with doxorubicin. However, results revealed that antibody conjugated agents had diverse effects on the cell lines. At the optimized doses, conjugated antibodies, especially antibody-conjugated-antigen associated with active serum, destroy the MCF7 cells significantly in contrast to the little destruction of the HUVEC cells.

Initially, MCF7 and HUVEC cells were treated with various concentrations of the human serum; according to the results, a significant toxic effect occurred in the presence of higher than 40% of human serum; this border line was selected for the following tests. The unconjugated and conjugated anti MCF7 antibodies have significant effects on MCF7 cell viability unlike the safe effects on HUVEC cells. The antibody concentrations higher than 4 μg/mL could not elevate cytotoxicity, significantly in compression to lower doses (Figs 3–5).

The used serums stimulate the cytotoxicity of the antibody conjugated to the *E. coli* antigens, significantly. Also, utilization of the antigen conjugated antibody higher than 4 μg/mL,

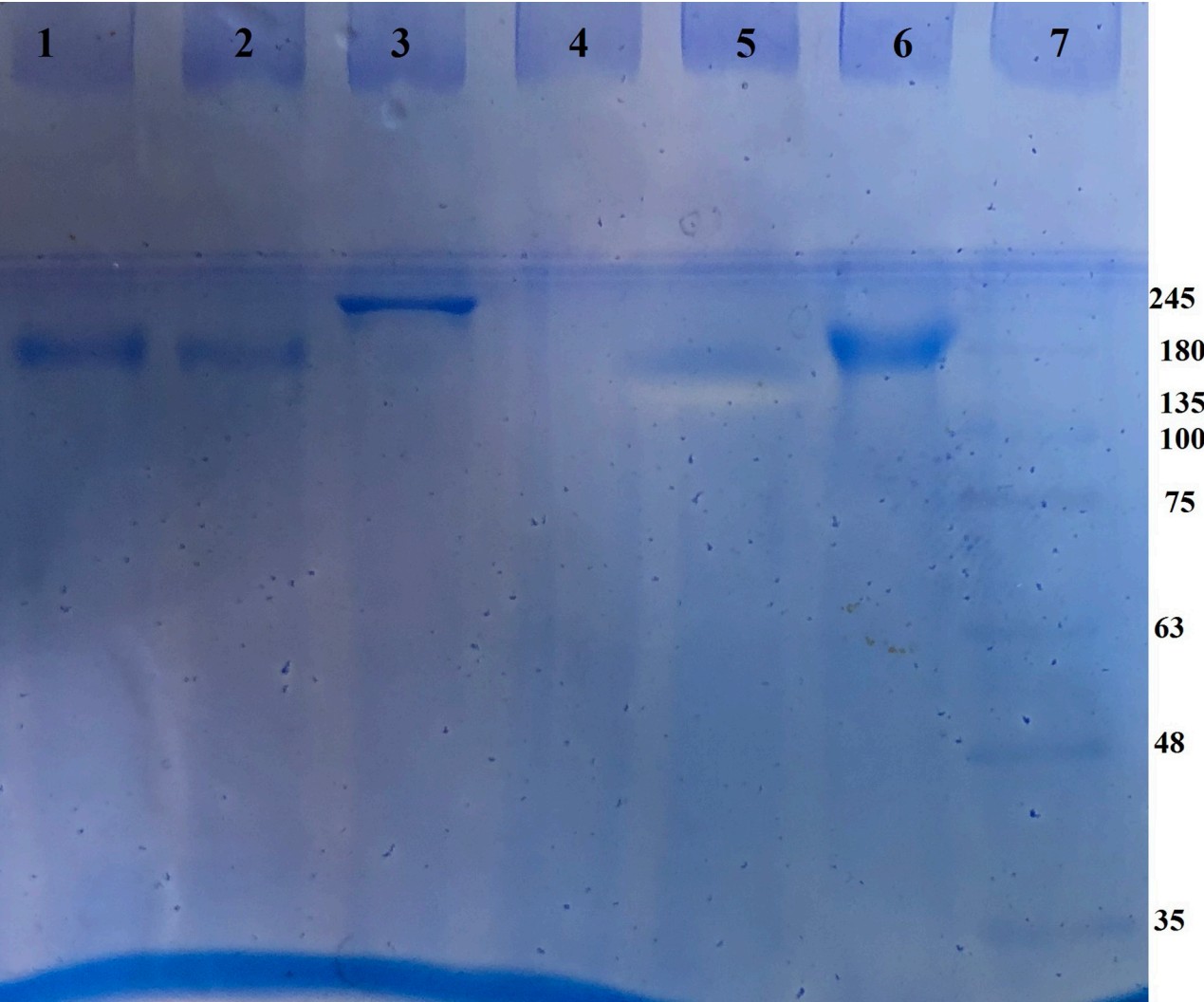

**Fig 1. The SDS-PAGE analysis of the antibody conjugation with *E. coli* antigens and doxorubicin.** The anti-MCF7 antibody (lane 1 and 2), antibody conjugated to *E. coli* antigens (lane 2), blank (lane 4 and 5) and antibody conjugated to doxorubicin (lane 6); lane 7 shows the protein molecular weight ladder.

resulted in a significant cytotoxic effect on the HUVEC cell line; the obtained results are shown in Figs 3 and 6.

Findings revealed that doxorubicin had significant dose-dependently cytotoxic effects on the entire examined dose on both MCF7 and HUVEC cells. The used doxorubicin concentrations appeared approximately equal or higher destruction levels of the HUVEC than MCF7 cell line. The doxorubicin conjugated to a specific antibody at a concentration ranging to 1–5 μg showed a significantly higher cytotoxic effect on MCF7 than HUVEC cells; (Figs 7–9).

## In vivo triggering of inflammation

The scanning electron microscope (SEM) confirmed the homogeneous distribution of the cells into scaffolds (Fig 10). After washing, the prepared scaffolds were put into the rat peritoneal cavity. After a one week treatment period, the scaffolds were removed and the inflammatory and the fibroblast cell number/area ratios were counted.

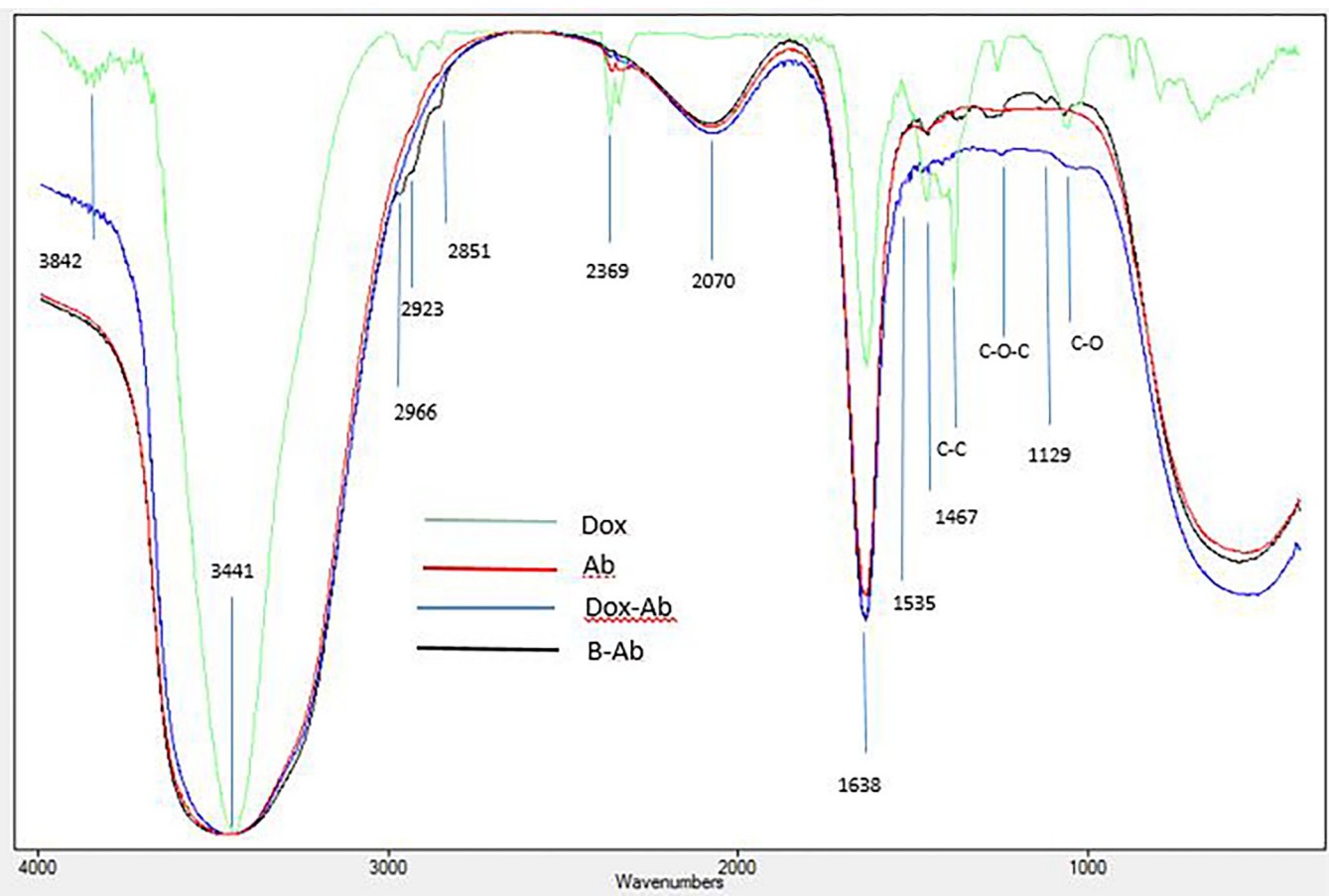

**Fig 2. The Attenuated Total Reflection Fourier-Transform Infrared Spectroscopy (ATR-FTIR) analysis of the antibodies conjugated to doxorubicin (Ab-Dox) and bacterial antigens (B-Ab), purified anti-MCF7 antibodies (Ab) and doxorubicin (Dox).**

The antibody conjugated to *E. coli* antigens significantly stimulated the inflammatory immune responses against the grafted MCF7 scaffolds; however, these responses were significantly suppressed in the treating mice with an antibody conjugated to doxorubicin (Table 1 and Fig 11).

## Discussion

Tumor niches always have immunosuppressive properties. This research provided a new approach to trigger the host innate immune system against tumor cells. Based on the facts, the innate immune cells recognize the bacterial antigens by using pattern recognition receptors; in addition, the innate immune molecules such as complement, mannose binding lectin, C- reactive protein can recognize the bacterial antigen. It is important to mention that people always have poly-reactive anti-bacterial titters against environmental bacteria such as *Escherichia coli*, *staphylococcus aureus*. This existing or induced antibacterial titer could bind to the bacterial antigens conjugated to the anti-cancer antibody. Based on these hypotheses, the current research evaluated the effects of the [antibody-bacterial antigens] complex on MCF7 cells and a normal cell line of HUVEC. The in vitro results indicated that the active serum destroyed the tumor cell lines in the presence of the [antibody-bacterial antigens]. The [antibody-bacterial

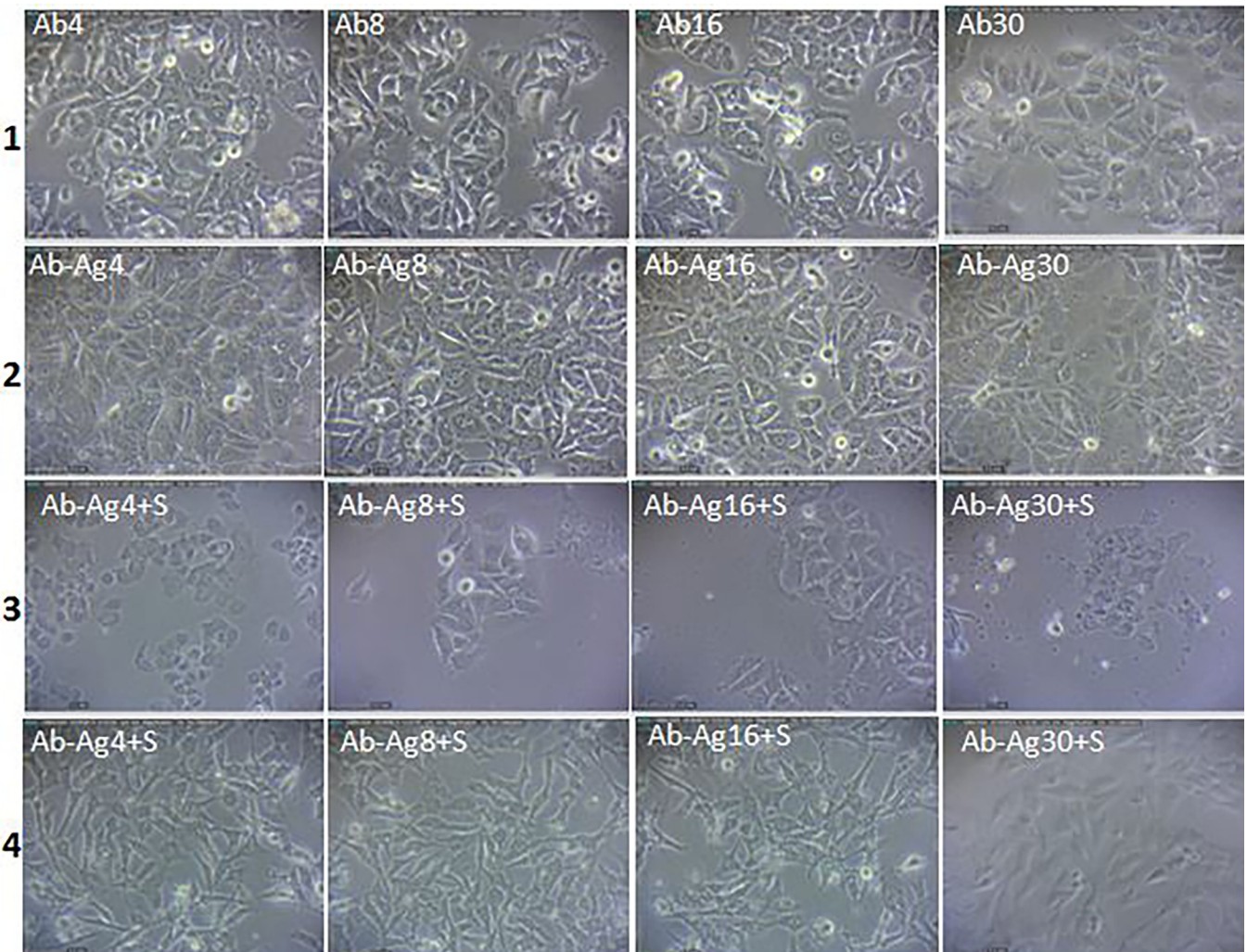

**Fig 3. The effects of different concentrations (4, 8, 16 and 30μg) of the specific antibody (Ab), antibody conjugated to *E. coli* antigens (Ab-Ag) and antibody conjugated to *E. coli* antigens associated with active serum (Ab-Ag+S) on morphology of MCF7 (row 1, 2 and 3) and HUVEC cell lines (row 4).**

antigens], contrary to other treatments, have negligible effects on the normal laboratory cell model [23], the HUVEC cell line.

Doxorubicin has functional groups of CO, OH, H and NH2 that created the related specific peaks blew 1500 cm$^{-1}$ at 1076, 1285, 1409 cm$^{-1}$ for C-O, C-O-C and C-C, respectively. These peaks were missed in the antibody-Dox complex. The carbonyl (–HC O) stretching bands were observed at 1640 cm$^{-1}$. Also, quinone and ketone carbonyl groups created FTIR bands at 3432, 2925, 1725, 1623, 1407 and 1008 cm$^{-1}$ [24, 25]. These reports are in accordance with current study. Antibody molecules contain C, O, N, and S; so, the complex which contained antibody shows a higher concentration of these elements [26]. The used antibody showed the expected peaks of the amine I groups at 1638, amine II at 1467 and the carboxyl groups at 3441. The new appearing peaks in the regions 1300–2000 cm$^{-1}$ could be created in the presence of the protein peaks of the bacterium [27]. In the region 1402–1457 cm$^{-1}$, bands due to carbohydrate, glycoprotein and lipid peaks should be observed at 1402–1457 cm$^{-1}$ [28, 29].

There are different approaches for active and passive immunotherapy of the various cancer types. Tumor cells have specific and common epitopes for B and T lymphocytes, which are

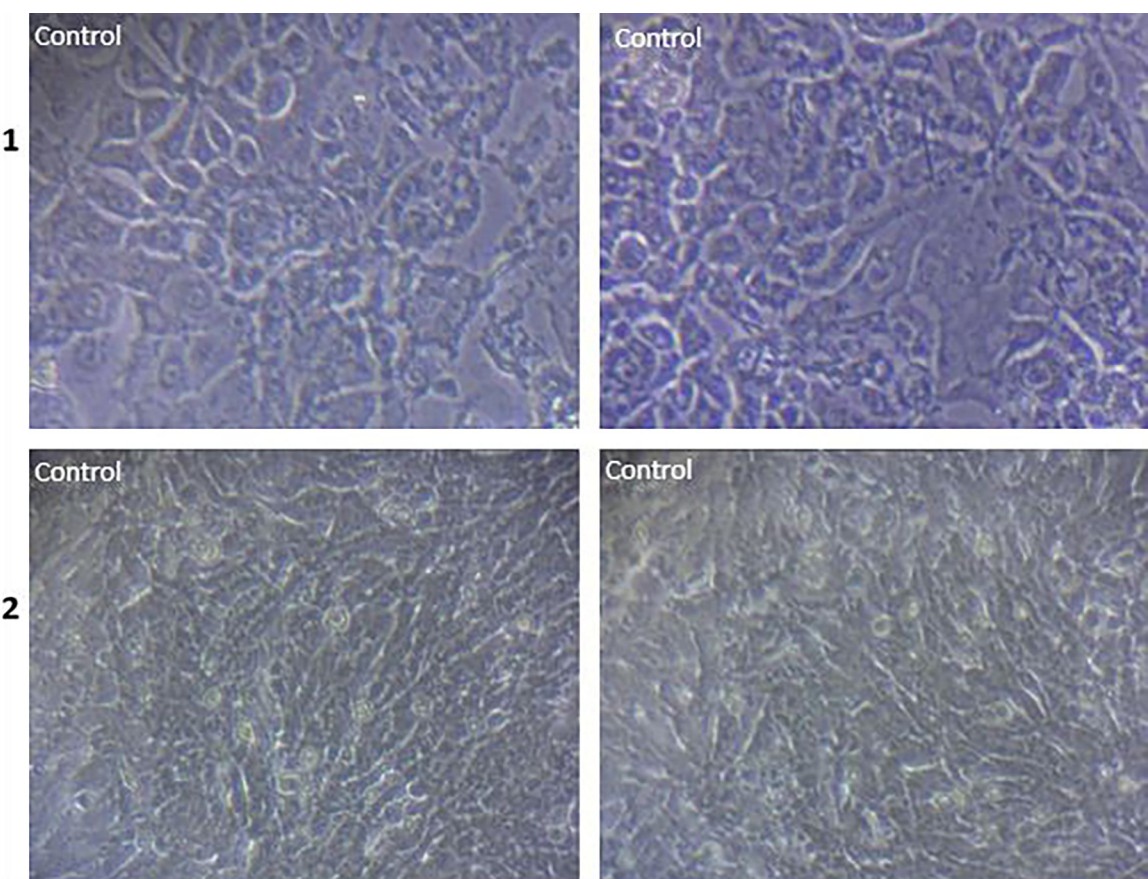

**Fig 4. Morphology of the control cell line of MCF7 (row 1) and HUVEC cell lines (row 2).**

targets for the immunotherapy. Active immunization is done by injection of the irradiated autologous or allogeneic tumor cells preparing tumor cell lysate or preparing dendritic cells which present tumor antigens [30]. Passive immunotherapies are transferred by utilization of the antibodies and adoptive cell therapy. In addition to the direct cytotoxic effects of the anti-cancer antibody, conjugation of the antibodies with radionuclides, drugs and toxins provides a variety of FDA approved treatments. The current study presented a combined route of active and passive immunotherapy against cancer cells. The covalent conjugation of the drug to its carrier is a certain approach to preparing the target therapy agents. In contrast to conventional cancer treatment, current research has delivered the antigens of the pathogenic *E. coli* to the cancer cells; in this respect, the efficacy of the conjugated agent has been confirmed in vivo and in vitro experiments.

Previously, the roles of antibodies were defined as blocking of the surface receptors and survival pathways. In parallel, the innate immune cells have Fc receptors; so, the targeted cells are marked by antibodies [31]. Two pathways of antibody dependent cell cytotoxicity (ADCC) and antibody-dependent cellular phagocytosis (ADCP) were revealed at this point. Recently, researchers and clinicians have tried to combine chemotherapy and radiation therapy with antibody target therapy since they believe it could combat with iatrogenic and tumor immuno-suppressive effects; these regimens increase the survival rate of the patients [32]. Higher affinity of the FcγRIIIA towards anti-cancer IgG has been achieved by modification of the Fc domain structure or Fc domain oligosaccharide content, which has resulted in higher ADCC

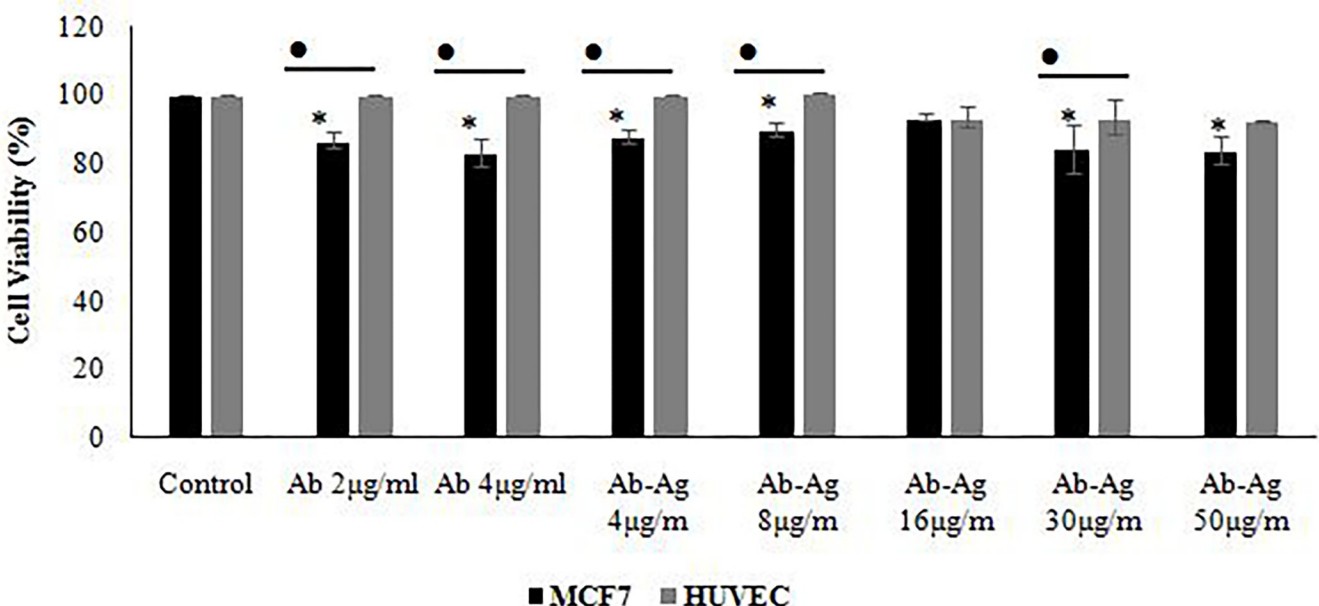

**Fig 5. Viability of the MCF7 and HUVEC cell lines after treatment with antibody and antibody conjugated to *E. coli* antigens using MTT test.** The significant differences in the treatment compared to control group and MCF7 compared to HUVEC were shown by using asterisks (*) and the bullet (•), respectively ($p<0.05$).

responses of natural killer cells [33–35]. The anti-cancer antibodies always stimulate both ADCC and CMC; however, anti-tumor activity of the antibodies such as alemtuzumab could be achieved through only activation of CMC [36]. The current conjugated antibody delivers the bacterial antigens to the cancer cell surface; the event could stimulate various aspects of the innate and adaptive immune responses, as result of the in vivo and in vitro experiments.

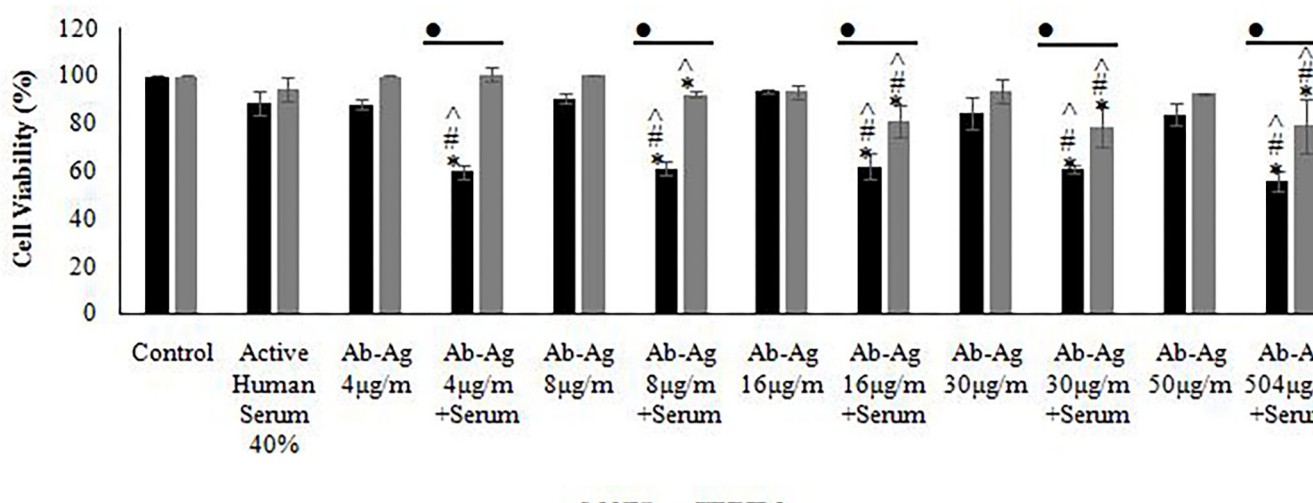

**Fig 6. Viability of the MCF7 and HUVEC cell lines after treatment with active serum, antibody conjugated to *E. coli* antigens and antibody conjugated to *E. coli* antigens associated with active serum using MTT test.** The significant difference in the treatment compared to control group, MCF7 compared to HUVEC, treatment compared to active human serum group and Ab-Ag treatment compared to parallel dose without active serum was shown by using asterisks (*), the bullet (•), hashtag (#) and wedge (^), respectively ($p<0.05$).

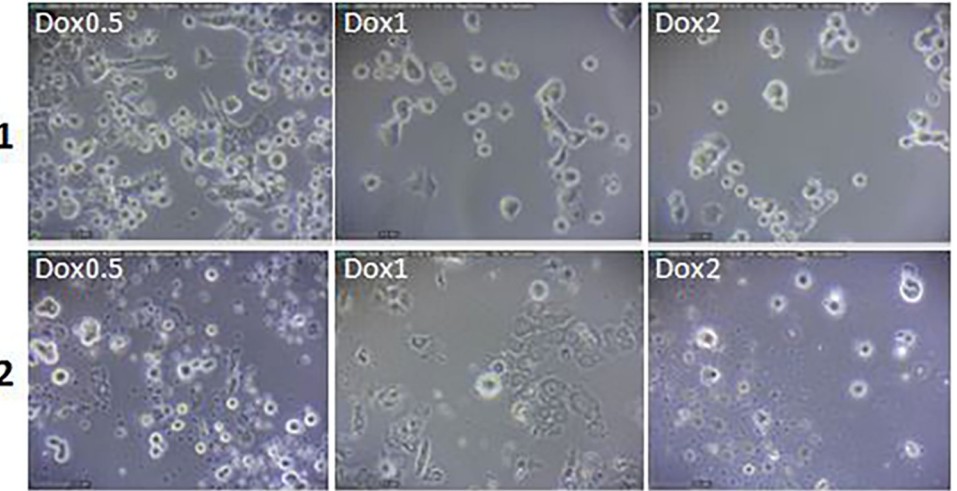

**Fig 7. The effects of different concentrations (0.5, 1 and 2µg) of the doxorubicin (Dox) on morphology of the MCF7 (row 1) and HUVEC cell lines (row 2).**

The complement mediated cytotoxicity (CMC) is another characteristic of the immune-related response against the targeted cells [3]. Also, the enhancement of the complement activation through therapeutic anticancer antibody has been obtained through Ofatumumab in chronic lymphocytic leukemia. There is always a direct relationship between complement activation and therapeutic effects of the antibodies [3, 37].

The anticancer activity of the bacteria went back to a century ago, when it was revealed that *Streptococcus pyogenes* limited the growth of certain tumor types in infected patients [38]. Also, William Coley observed the curative effects of erysipelas in treatment of end stage cancers [39]. As a delivery agent and vector, the cancer antigens were introduced to the immune system by using facultative intracellular bacteria, *Listeria monocytogenes*. This approach stimulates potent innate and cell mediated immunity against all tumor types such as cervical cancer [40].

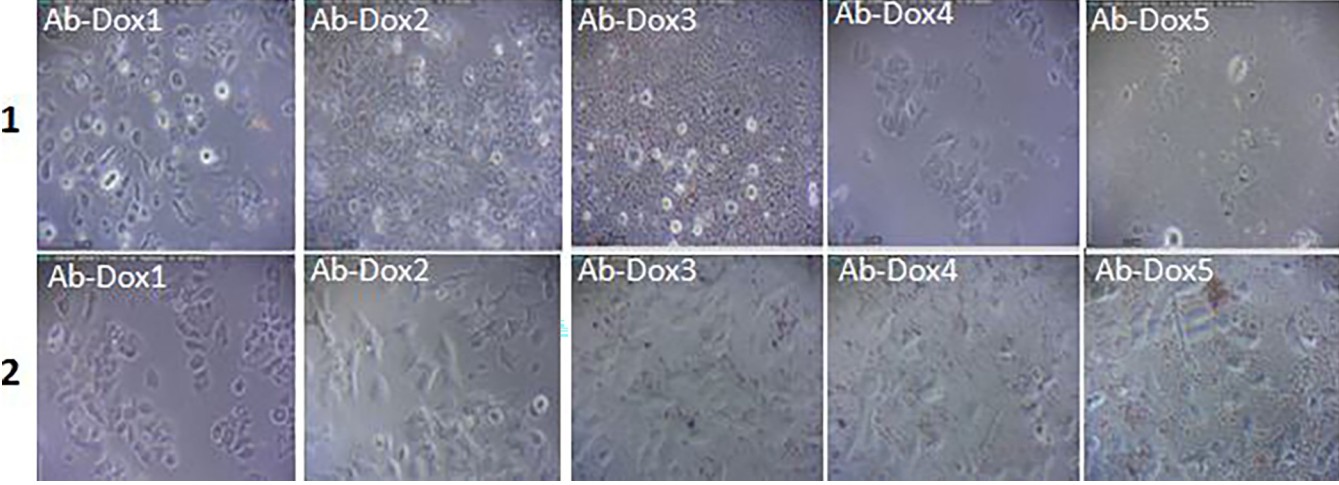

**Fig 8. The effects of different concentrations (1, 2, 3, 4 and 5µg) of the antibody conjugated to doxorubicin (Ab-Dox) on morphology and the availability of the MCF7 (row 1) and HUVEC cell lines (row 2).**

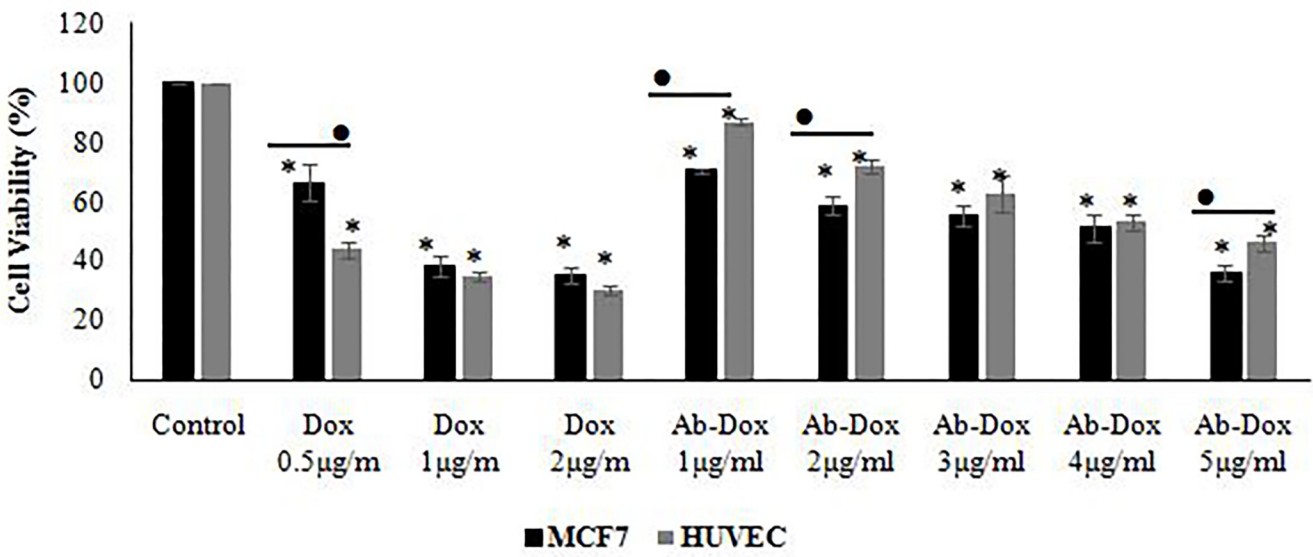

**Fig 9. Viability of the MCF7 and HUVEC cell lines after treatment with doxorubicin and antibody conjugated to doxorubicin using the MTT test.** The significant differences in the treatment compared to control group and MCF7 compared to HUVEC were shown by using asterisks (*) and the bullet (•), respectively (p<0.05).

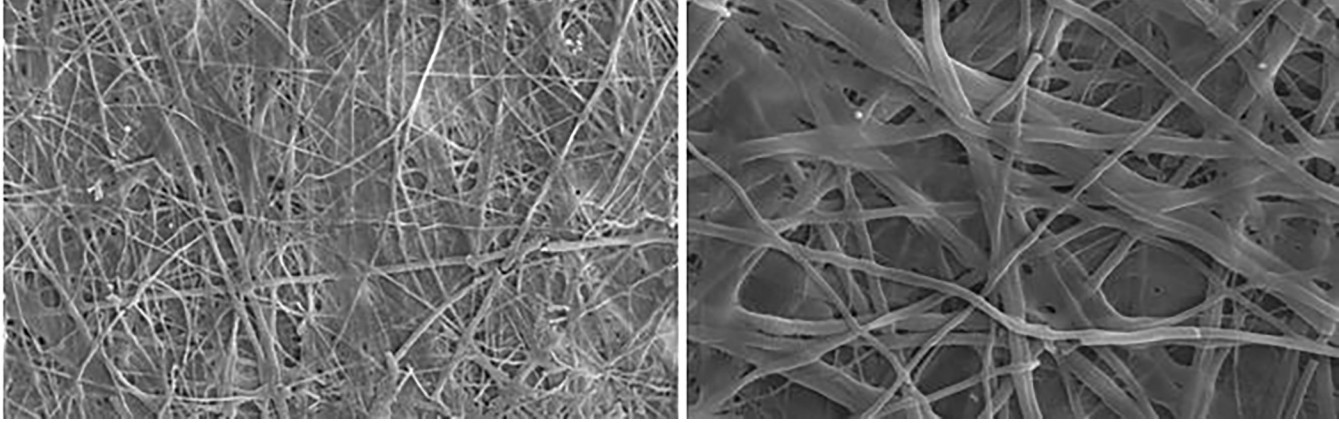

**Fig 10. The scanning electron microscope (SEM) of MCF7 adhesion and spreading on polylactic acid scaffolds during 6 days of incubation.**

**Table 1. The number of the inflammatory and fibroblast cells in grafted MCF7 scaffolds.**

|  | Inflammatory | Fibroblasts | Inflammatory/Fibroblasts |
|---|---|---|---|
| Control | 13.6±4.30[a] | 21.7±7.89[a] | 0.76±0.95[a] |
| Ab-Ag | 24.54±6.91[b] | 6.18±2.48[b] | 4.87±1.77[b] |
| Ab-Dox | 15.33±5.44[a] | 14.61±6.85[c] | 1.16±0.52[a] |

The MCF7 nanofiber scaffold was grafted into three groups of rats including control, Ab-Ag and Ab-Dox; the groups were treated with 100 µL of PBS, antibody conjugated to *E. coli* antigens (10 µg/mL) and antibody conjugated to doxorubicin (10 µg/mL) for seven days, respectively. Letter in each column indicate the significant difference between the groups (p<0.05).

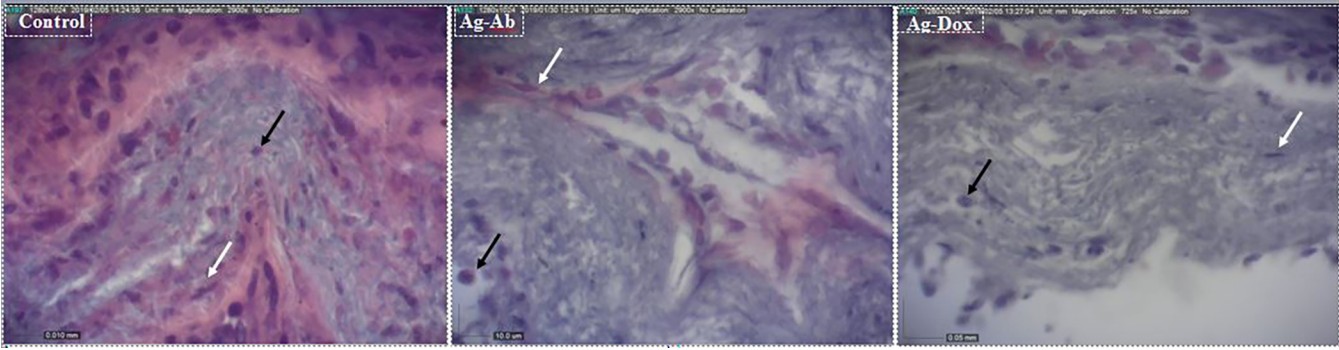

**Fig 11. Histological changes to the grafted MCF7 scaffolds; recruitment of the inflammatory (black arrows) and fibroblast cells (white arrows) occurred at different proportions.**

By using recombinant *Salmonella typhimurium*, another study triggered the antitumor response without noticeable toxicity and accelerated the rejection of the primary tumors [41]. Also, attenuated *S. typhimurium* could infect malignant cells and direct the immune response against tumor cells [42]. The bacterial cell wall antigen, Bacillus Calmette-Guerin, is another well-known agent that is used as an adjuvant immunotherapy [43]. The bacterial toxins and endotoxins were also tested as anticancer therapy; these elements could destroy the cancer cells, cancer vaccine and production of the immunotoxins [44]. Production of the inflammatory cytokines was induced by systemic injection of the *Clostridium novyi-NT* spores; the attraction of the inflammatory cells including, in chronological order, neutrophils, monocytes and lymphocytes elicited the anticancer responses [45]. In addition, the inflammatory reactions resulted in the cellular immune responses, production of the reactive oxygen species and proteolysis enzymes [16]. This study, like other related research, confirmed the utilization of nonpathogenic bacteria as a novel immunotherapeutic agent.

## Conclusion

This experiment revealed that the antibody conjugated-bacterial antigens triggered the innate immune response including cytotoxic effects of the serum components and inflammatory responses against the MCF7 cancer cell line. This anti-cancer agent at the optimal dose had negligible effects on normal cell line. However, this therapy method needs a complementary drug delivery approach for masking the antigens and transfer to the tumor sites to inhibit stimulation of the immune responses in other sites of the body.

## Supporting information

**S1 File.**
(XLSX)

**S2 File.**
(XLSX)

**S3 File.**
(DOCX)

**S4 File.**
(RAR)

**S1 Raw images.**
(PDF)

## Acknowledgments

Authors thank Dr Hadi Imani Rastabi for her kind cooperation in surgical operations.

## Author Contributions

**Conceptualization:** Mohammad Khosravi.

**Data curation:** Mohammad Khosravi, Fatemeh KhademiMoghadam.

**Formal analysis:** Mohammad Khosravi.

**Investigation:** Mohammad Khosravi, Kaveh Khazaeil, Fatemeh KhademiMoghadam.

**Methodology:** Mohammad Khosravi, Kaveh Khazaeil, Fatemeh KhademiMoghadam.

**Supervision:** Mohammad Khosravi.

**Validation:** Mohammad Khosravi, Kaveh Khazaeil, Fatemeh KhademiMoghadam.

**Visualization:** Mohammad Khosravi, Kaveh Khazaeil.

**Writing – original draft:** Mohammad Khosravi.

**Writing – review & editing:** Mohammad Khosravi.

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
