## [Decision Letter · Decision Letter 0]

2 Aug 2022

PONE-D-22-18049Triggering of the in vitro and in vivo immune response against MCF7 tumor cell line using conjugated antibody with bacterial antigensPLOS ONE

Dear Dr. Khosravi,

Thank you for submitting your manuscript to PLOS ONE. After careful consideration, we feel that it has merit but does not fully meet PLOS ONE’s publication criteria as it currently stands. Therefore, we invite you to submit a revised version of the manuscript that addresses the points raised during the review process.

We look forward to receiving your revised manuscript.

Kind regards,

Amitava Mukherjee, ME, Ph.D.

Academic Editor

PLOS ONE

Journal Requirements:

"This study was financially supported by Shahid Chamran University of Ahvaz, Ahvaz, Iran, under the grant number of 98/3/05/14909. "

"The funders had no role in study design, data collection and analysis, decision to publish, or preparation of the manuscript"

6. PLOS ONE now requires that submissions reporting blots or gels include original, uncropped blot/gel image data as a supplement or in a public repository. This is in addition to complying with our image preparation guidelines described at https://journals.plos.org/plosone/s/figures#loc-blot-and-gel-reporting-requirements. These requirements apply both to the main figures and to cropped blot/gel images included in Supporting Information. If the manuscript is positively reviewed, we will ask the authors to provide any missing raw image data for blot/gel results when they submit their first revision. As part of your review, please ensure that figures reporting blot or gel images comply with the journal’s image preparation guidelines and that the original data are provided following the journal’s request.  If you have any questions or concerns about blot/gel figures or data for this submission, please email us at plosone@plos.org before issuing a decision letter.

7. Please amend your list of authors on the manuscript to ensure that each author is linked to an affiliation. Authors’ affiliations should reflect the institution where the work was done (if authors moved subsequently, you can also list the new affiliation stating “current affiliation:….” as necessary).

8. Please amend either the abstract on the online submission form (via Edit Submission) or the abstract in the manuscript so that they are identical.

Reviewers' comments:

Reviewer's Responses to Questions

**Comments to the Author**

1. Is the manuscript technically sound, and do the data support the conclusions?

Reviewer #1: Yes

2. Has the statistical analysis been performed appropriately and rigorously? 

Reviewer #1: Yes

3. Have the authors made all data underlying the findings in their manuscript fully available?

Reviewer #1: Yes

4. Is the manuscript presented in an intelligible fashion and written in standard English?

Reviewer #1: No

5. Review Comments to the Author

Reviewer #1: RE: Triggering of the in vitro and in vivo immune response against MCF7 tumor cell line using conjugated antibody with bacterial antigens

1- Please do change the manuscript tittle to: Triggering of the immune response against MCF7 cell line using conjugated antibody with bacterial antigens: In-vitro and In-vivo study

2- In general, there are some typo errors and grammar mistake. Please make sure about them.

3-Please rewrite the main finding in abstract and conclusion sections to be clear and readable for readers in future. (Do focus on novelty of your work)

4-The Introduction section need to improve. In introduction, you have to write sufficient background information, and the purpose of the article is clearly defined at the end of the introduction part.

5- Most of materials and methods section needed to add references.

So, do cite the following articles in missing place.

• Al Salman, H. N. K., Ali, E. T., Jabir, M., Sulaiman, G. M., and Al Jadaan, S. A. (2020). 2 Benzhydrylsulfinyl N hydroxyacetamide Na extracted from fig as a novel cytotoxic and apoptosis inducer in SKOV 3 and AMJ 13 cell lines via P53 and caspase 8 pathway.‏ .‏ Eurpean food research technology. And Al-Ziaydi, A. G., Al-Shammari, A. M., Hamzah, M. I., Kadhim, H. S., and Jabir, M. S. (2020). Newcastle disease virus suppress glycolysis pathway and induce breast cancer cells death. VirusDisease As a refrences for cell cytotoxicity in vitro.

• Ali, I. H., Jabir, M. S., Al-Shmgani, H. S., Sulaiman, G. M., & Sadoon, A. H. (2018, May). Pathological And immunological study on infection with escherichia coli in ale balb/c mice. In Journal of Physics: Conference Series (Vol. 1003, No. 1, p. 012009). IOP Publishing.). As areference for Hand E staining method.

• Younus, A., Al-Ahmer, S., and Jabir, M. (2019). Evaluation of some immunological markers in children with bacterial meningitis caused by Streptococcus pneumoniae. Research Journal of Biotechnology, 14, 131-133. As a reference for statistical analysis.

6- In figures legends of figure 2 you wrote: Figure 2. Figure 1. The Attenuated Total Reflection Fourier-Transform Infrared 537 Spectroscopy (ATR-FTIR) analysis of the antibodies conjugated to doxorubicin (Ab-Dox) 538 and bacterial antigens (B-Ab), purified anti-MCF7 antibodies (Ab) and doxorubicin (Dox). Please do remove Figure 1 from sentence.

7- Please do change figure 4 legend from The control cell line of MCF7 (row 1) and HUVEC cell lines (row 2). To Morphology of the control cell line of MCF7 (row 1) and HUVEC cell lines (row 2).

8- Please do change figure 11 legend from. Histological slides of the grafted MCF7 scaffolds; recruitment of the 567 inflammatory (black arrows) and fibroblast cells (white arrows) occurred at different 568 proportions. To Histological changes of the grafted MCF7 scaffolds; recruitment of the 567 inflammatory (black arrows) and fibroblast cells (white arrows) occurred at different 568 proportions.

6. PLOS authors have the option to publish the peer review history of their article (what does this mean?). If published, this will include your full peer review and any attached files.

Reviewer #1: **Yes: **Majid Jabir

---

## [Author Response · Author response to Decision Letter 0]

27 Aug 2022

Responses to reviewers' comments

1. Please do change the manuscript tittle to: Triggering of the immune response against MCF7 cell line using conjugated antibody with bacterial antigens: In-vitro and In-vivo study

Thanks for your suggestion; the tittle replaced as you noted.

2. In general, there are some typo errors and grammar mistake. Please make sure about them.

The language of the manuscript checked for grammar and dictation mistakes; the corrected words and sentences highlighted in red. Thanks for your valuable comments.

3. Please rewrite the main finding in abstract and conclusion sections to be clear and readable for readers in future. (Do focus on novelty of your work)

The main finding in abstract and conclusion sections were revised as you noted about novelty of the research.

4. The Introduction section need to improve. In introduction, you have to write sufficient background information, and the purpose of the article is clearly defined at the end of the introduction part.

The introduction section was revised according to the comment.

5. Most of materials and methods section needed to add references.

The references were added to the methods. The new references was highlighted in red.

6. In figures legends of figure 2 you wrote: Figure 2. Figure 1. The Attenuated Total Reflection Fourier-Transform Infrared 537 Spectroscopy (ATR-FTIR) analysis of the antibodies conjugated to doxorubicin (Ab-Dox) 538 and bacterial antigens (B-Ab), purified anti-MCF7 antibodies (Ab) and doxorubicin (Dox). Please do remove Figure 1 from sentence.

This mistake was corrected. Thanks for your valuable comments.

7. Please do change figure 4 legend from The control cell line of MCF7 (row 1) and HUVEC cell lines (row 2). To Morphology of the control cell line of MCF7 (row 1) and HUVEC cell lines (row 2). 

The legend was corrected as mentioned. Thanks for your valuable comments.

8. Please do change figure 11 legend from Histological slides of the grafted MCF7 scaffolds; recruitment of the inflammatory (black arrows) and fibroblast cells (white arrows) occurred at different proportions. To Histological changes of the grafted MCF7 scaffolds; recruitment of the inflammatory (black arrows) and fibroblast cells (white arrows) occurred at different proportions.

The legend was corrected as mentioned. Thanks for your valuable comments.

---

## [Decision Letter · Decision Letter 1]

26 Sep 2022

Triggering of the immune response to MCF7 cell line using conjugated antibody with bacterial antigens: In-vitro and In-vivo study

PONE-D-22-18049R1

Dear Dr. Khosravi,

We’re pleased to inform you that your manuscript has been judged scientifically suitable for publication and will be formally accepted for publication once it meets all outstanding technical requirements.

Kind regards,

Amitava Mukherjee, ME, Ph.D.

Academic Editor

PLOS ONE

Additional Editor Comments (optional):

Reviewers' comments:

Reviewer's Responses to Questions

**Comments to the Author**

1. If the authors have adequately addressed your comments raised in a previous round of review and you feel that this manuscript is now acceptable for publication, you may indicate that here to bypass the “Comments to the Author” section, enter your conflict of interest statement in the “Confidential to Editor” section, and submit your "Accept" recommendation.

Reviewer #1: All comments have been addressed

2. Is the manuscript technically sound, and do the data support the conclusions?

Reviewer #1: Yes

3. Has the statistical analysis been performed appropriately and rigorously? 

Reviewer #1: Yes

4. Have the authors made all data underlying the findings in their manuscript fully available?

Reviewer #1: Yes

5. Is the manuscript presented in an intelligible fashion and written in standard English?

Reviewer #1: Yes

6. Review Comments to the Author

Reviewer #1: Thanks for your efforts, and response. The revised version is improved, looking very good, and interested.

7. PLOS authors have the option to publish the peer review history of their article (what does this mean?). If published, this will include your full peer review and any attached files.

Reviewer #1: No

---

## [Editor Report · Acceptance letter]

28 Sep 2022

PONE-D-22-18049R1 

Triggering of the immune response to MCF7 cell line using conjugated antibody with bacterial antigens: In-vitro and In-vivo study 

Dear Dr. Khosravi:

I'm pleased to inform you that your manuscript has been deemed suitable for publication in PLOS ONE. Congratulations! Your manuscript is now with our production department. 

Kind regards, 

on behalf of

Professor Dr. Amitava Mukherjee 

Academic Editor

PLOS ONE